# The potential for diversion of prescribed opioids among orthopaedic patients: Results of an anonymous patient survey

Kala Sundararajan[1,2], Prabjit Ajrawat[3], Mayilee Canizares[1,2], J. Denise Power[1,2], Anthony V. Perruccio[1,2,3,4], Angela Sarro[1,3], Luis Montoya[1,2], Y. Raja Rampersaud[1,2,3]*, the University Health Network Division of Orthopaedic Surgery[¶]

1 Division of Orthopaedic Surgery, Schroeder Arthritis Institute, University Health Network, Toronto, Ontario, Canada, 2 Krembil Research Institute, University Health Network, Toronto, Ontario, Canada, 3 Department of Surgery, Division of Orthopaedics, University of Toronto, Toronto, Ontario, Canada, 4 Institute of Health Policy, Management & Evaluation, Dalla Lana School of Public Health, University of Toronto, Toronto, Ontario, Canada

¶ Membership of the University Health Network Division of Orthopaedic Surgery is provided in the Acknowledgments
* Raja.Rampersaud@uhn.ca

**Data Availability Statement:** The data underlying the findings of this study contain sensitive patient information and cannot be shared publicly, per our institutional research ethics board (REB). Readers

## Abstract

### Introduction

Diversion of prescription opioid medication is a contributor to the opioid epidemic. Safe handling practices can reduce the risk of diversion. We aimed to understand: 1) if orthopaedic patients received instructions on how to safely handle opioids, 2) their typical storage/disposal practices, and 3) their willingness to participate in an opioid disposal program (ODP).

### Methods

Cross-sectional study of adult orthopaedic patients who completed an anonymous survey on current or past prescription opioid use, instruction on handling, storage and disposal practices, presence of children in the household, and willingness to participate in an ODP. Frequencies and percentages of responses were computed, both overall and stratified by possession of unused opioids.

### Results

569 respondents who reported either current or past prescription opioid use were analyzed. 44% reported receiving storage instructions and 56% reported receiving disposal instructions from a health care provider. Many respondents indicated unsafe handling practices: possessing unused opioids (34%), using unsafe storage methods (90%), and using unsafe disposal methods (34%). Respondents with unused opioids were less likely to report receiving handling instructions or using safe handling methods, and 47% of this group reported having minors or young adults in the household. Respondents who received storage and disposal instructions were more likely to report safe storage and disposal methods. Seventy-four percent of respondents reported that they would participate in an ODP.

can request access to the data by completing a data request following a research proposal and a data access agreement approved by the University Health Network (UHN) Arthritis Program Research Committee and the UHN REB, signed by all parties. The UHN Arthritis Program Research Committee can be contacted via Christian Veillette at Christian. Veillette@uhn.ca. The UHN REB can be contacted at: University Health Network Research Ethics Board, 700 University Avenue, 4th Floor, Toronto, Ontario M5G 1Z5, Canada (e-mail: reb@uhnresearch.ca, phone: 1-416-581-7849).

**Funding:** This study was funded by the Toronto General and Western Hospital Foundation (TGWHF, https://uhnfoundation.ca/). TGHWF had no role in study design, data collection and analysis, decision to publish, or preparation of the manuscript.

**Competing interests:** I have read the journal's policy and the authors of this manuscript have the following competing interests: Y. Raja Rampersaud has received royalties from Medtronic and holds investments in Arthur Health Corporation. This does not alter our adherence to PLOS ONE policies on sharing data and materials. All other co-authors have no competing interests to report.

## Conclusion

While many orthopaedic patients report inadequate education on safe opioid handling and using unsafe handling practices, findings suggest targeted education is associated with better behaviours. However, patients are willing to safely dispose of unused medication if provided a convenient option. These findings suggest a need to address patient knowledge and behavior regarding opioid handling to reduce the risk of opioid diversion.

## Introduction

The opioid epidemic is a critical public health issue in Canada [1]. Opioid-related hospitalizations and rates of opioid poisoning emergency department (ED) visits continue to increase [2]. Rates of hospitalization for opioid poisoning are growing among youth (age 15 to 24) and younger adults (age 25 to 44), with respective increases of 53% and 62% from 2013 to 2017 [3]. In 2019, one in ten Ontario students in intermediate or secondary school reported using prescription opioids not prescribed to them, with the majority reporting that they obtained these drugs from their home [4]. However, opioid diversion in the household has received little attention from the medical community despite it being a significant source of opioid misuse in this vulnerable group.

Opioids are commonly prescribed for musculoskeletal pain and post-surgical pain [5–7]. Among physicians, orthopaedic surgeons are the third highest group of opioid prescribers [8,9]. Consequently, orthopaedic surgery patients may be important contributors to opioid diversion, as the majority of surgical patients use less than half of their prescribed opioid medication and often keep the surplus [8,9]. While current efforts to address over-prescription of opioids are a critical step towards reduction of opioid diversion [10,11], little is known about how opioid handling in the household may relate to diversion. The primary objectives of this study were to develop a better understanding of: 1) whether orthopaedic patients receive instruction on how to safely handle opioids, 2) their typical storage and disposal practices, and 3) their willingness to participate in a hypothetical hospital-based opioid disposal program. Secondarily, we examined how these factors varied among patients with and without unused opioids at home, and whether patient responses varied with the presence of children or young adults in the home.

## Materials and methods

### Study design

Cross-sectional study of consecutive patients who visited the outpatient orthopaedic clinic at Toronto Western Hospital from May to July 2017. Patients were invited to complete and return an anonymous paper survey to an unmanned box in the clinic. Respondents who reported current or former use of opioids were included in the analysis sample.

### Data collection

The survey collected respondents' sex, age group, history of being prescribed opioids, previous instruction regarding safe opioid handling practices, actual handling practices, and the presence of children and/or young adults in their household (see S1 Appendix). Survey items were generated based on review of existing studies and Canadian recommendations [12,13].

## Survey measures

**Opioid use.**   Respondents were asked if they had ever been prescribed opioid/narcotic pain medications (options: never; prescribed in the past but not currently using; currently using sometimes but not daily; currently using daily). This question was not limited to prescription by an orthopaedic surgeon.

**Unused opioids at home.**   Respondents were asked if they had any opioids at home that were no longer being used or were expired ("Yes, prescribed to me", "Yes, prescribed to someone else", or "No"). Respondents reporting any unused opioids at home were collapsed into one group.

**Minors and/or young adults in the home.**   Respondents were asked if children, teenagers, or young adults lived in or visited their home ("No", "Yes, young children aged 0–6 years", "Yes, older children aged 7–12 years", "Yes, teenagers aged 13–17 years", and/or "Yes, young adults aged 18–25 years").

**Instruction regarding safe opioid storage/disposal.**   Respondents who had been prescribed opioids were asked if they had ever received information on how to a) store and b) dispose of opioids ("Yes, from a pharmacist", "Yes, from a healthcare provider", "Yes, from another source", "No, I have never received such information", and/or "I don't know"). Appropriate instruction was defined as receiving the information from a pharmacist or health care provider.

**Opioid storage.**   Respondents were asked how they typically stored opioids in their home ("Cabinet/storage with a latch", "Cabinet/storage with a lock", "Cabinet/storage with no latch or lock", or "Other"). Canadian guidelines state that opioids should be stored "out of sight and reach of children and pets" and "in a safe place to prevent theft, problematic use or accidental exposure" [12]. For this analysis, locked opioid storage locations were considered secure, in accordance with existing studies [14–17].

**Opioid disposal.**   Respondents were asked how they typically disposed of opioids ("Flush down the toilet or sink", "Throw away in garbage", "Mix with undesirable material [e.g. kitty litter, coffee grounds] and throw in garbage", "Return to a pharmacy or community take-back program", "I would not dispose of it", and/or "Other"). Safe opioid disposal was defined in accordance with Canadian guidelines[13] as either a) returning unused opioids to a pharmacy or community take-back program, or b) mixing the medication with undesirable material and throwing it in the garbage.

**Sharing opioid medication.**   Respondents were asked if they had ever shared opioids with another person ("Yes, I have shared opioid/narcotic medication prescribed to me", "Yes, I have used opioid/narcotic medication prescribed to someone else", and/or "No").

**Participation in a hospital-based take-back program.**   Respondents were asked if they would be willing to safely dispose of unused opioid medication (given that they had any) at their next hospital appointment ("Yes" or "No"). Respondents who were not willing to participate were asked to specify their reason(s), with options: "I am afraid my pain will come back and I'm not willing to be without it", "If I need it again, I will not be able to get it from a doctor in a timely manner", "I do not want to throw it away because I paid for it", "My friends or family members may need it", "I would prefer to dispose of it myself", and/or "Other".

## Statistical analysis

Responses were summarized with frequencies and percentages, both in the full sample and stratified by presence of unused opioid medication in the home. Respondents who reported any unused opioids at home were compared to respondents with no unused opioids using chi-square independence tests. Confidence intervals for differences in proportions were also

calculated using the Agresti and Coull method [18], with Bonferroni adjustment to control family-wise error when necessary [19]. A descriptive comparison of reported opioid storage method versus presence of children, teenagers, and/or young adults in the home versus was also performed. After confirming that all survey items had a completion rate of at least 90%, missing responses were removed using pairwise deletion.

### Ethics approval

This study was approved by the University Health Network Research Ethics Board (approval number 19–5808) as a retrospective analysis of data collected anonymously for a clinical quality initiative. Since identifying information was not collected, the informed consent requirement was waived.

## Results

Of 784 total respondents, 569 who indicated that they had been prescribed opioid medication in the past were included in the analysis sample. The remaining respondents either reported that they had never been prescribed opioids (n = 182) or did not indicate their use of opioids (n = 33). Sixty percent of the sample were women, and 58% were age 55 or older (Table 1). Seventy-two percent were former opioid users and 28% were current users. Forty-two percent reported at least one child, teenager, or young adult living in or visiting their home. Overall, 34% of respondents reported having unused opioids at home. Having unused opioids at home was not significantly related to age group, gender, or having minors/young adults at home, but was more common among former versus current opioid users (p<0.001).

Fewer than half of respondents reported that they received appropriate instruction in opioid medication handling: 43% received storage instructions and 37% received disposal instructions from a pharmacist or other healthcare provider (HCP) (Table 2). Many respondents stated that they received no storage (37%) or disposal (43%) instructions. Compared to respondents who reported no unused opioids at home, those who did have unused opioids were significantly less likely to have received storage instructions (36% vs. 46%, p = 0.035) or disposal instructions (27% vs. 43%, p<0.001) from a pharmacist or other HCP.

Many respondents reported unsafe opioid handling practices. Only 10% of respondents reported using a secure (locked) storage location; 90% reported an unlocked storage location. While 56% said they typically used a Health Canada-recommended disposal method [12] (i.e., return to the pharmacy or a take-back program), 44% reported neither of the recommended methods, with 14% reporting that they would not dispose of unused opioid medication. Eight percent reported sharing opioid medication (i.e., letting others use opioids prescribed to them and/or using opioids prescribed to someone else). Compared to respondents with no unused opioids, those with unused opioids at home were more likely to report unsafe handling practices, including storing opioids in an unsecured location (96% vs. 87% using unlocked storage, p = 0.001); disposing of opioids by throwing them in the garbage (23% vs. 12%, p = 0.001); stating that they would not dispose of opioids at all (21% vs 10%, p<0.001); and sharing prescription opioids (14% vs. 5%, p<0.001). Respondents in our sample who received storage and disposal instructions from a pharmacist or health care provider did report safer handling behaviors: they were less likely to have unused opioids at home (p = 0.005), more likely to keep opioids in a locked location (p<0.001), and more likely to dispose of opioids at a pharmacy or take-back program (p<0.001) (see S1A–S1C Table for comparative analysis of those who reported receiving instruction versus those who did not).

Seventy-two percent of respondents indicated they would be willing to return unused opioids in a potential hospital-based take-back program. Respondents with unused opioids were

**Table 1. Characteristics of opioid-using respondents, overall and by presence of unused opioids at home.**

| Measure | Category | All opioid ever-users, % (count) N = 569 | Unused opioid medication at home | | | |
| --- | --- | --- | --- | --- | --- | --- |
| | | | No unused opioids, % (count) N = 374 | Any unused opioids, % (count) N = 195 | Difference [95% CI] | P |
| Age group[a,b] | 18–24 | 3.9% (22) | 3.5% (13) | 4.6% (9) | 1.1% [-3.5, 6.2] | 0.316 |
| | 25–34 | 11.1% (63) | 10.2% (38) | 12.8% (25) | 2.6% [-4.8, 10.4] | |
| | 35–44 | 10.7% (61) | 11.8% (44) | 8.7% (17) | -3.1% [-9.9, 4.1] | |
| | 45–54 | 16.7% (95) | 16.9% (63) | 16.4% (32) | -0.5% [-9.0, 8.4] | |
| | 55–64 | 27.8% (158) | 29.5% (110) | 24.6% (48) | -4.9% [-15.0, 5.5] | |
| | 65–74 | 21.1% (120) | 18.8% (70) | 25.6% (50) | 6.9% [-2.9, 16.8] | |
| | 75+ | 8.6% (49) | 9.4% (35) | 7.2% (14) | -2.2% [-8.4, 4.4] | |
| | Age group not reported | — (1) | — (1) | — (0) | | |
| Female sex[a] | — | 59.6% (334) | 58.9% (216) | 61.1% (118) | 2.3% [-6.3, 10.7] | 0.665 |
| | Sex not reported | — (9) | — (7) | — (2) | | |
| Prescription opioid medication use[b] | Former | 71.5% (407) | 65.2% (244) | 83.6% (163) | 18.3% [10.0, 26.2] | <0.001* |
| | Sometimes | 11.8% (67) | 14.4% (54) | 6.7% (13) | -7.8% [-13.3, -1.7] | |
| | Daily | 16.7% (95) | 20.3% (76) | 9.7% (19) | -10.6% [-17.0, -3.6] | |
| Do children, teenagers, or young adults live in your home or visit your home?[c] | Young children (age 0–6) | 12.3% (70) | 10.7% (40) | 15.4% (30) | 4.7% [-1.2, 10.8] | 0.138 |
| | Older children (age 7–12) | 13.5% (77) | 14.4% (54) | 11.8% (23) | -2.6% [-8.2, 3.4] | 0.456 |
| | Teenagers (age 13–17) | 12.0% (68) | 9.9% (37) | 15.9% (31) | 6.0% [0.2, 12.1] | 0.050 |
| | Young adults (age 18–25) | 20.7% (118) | 19.8% (74) | 22.6% (44) | 2.8% [-4.2, 10.0] | 0.505 |
| | None of the above | 58.3% (332) | 61.2% (229) | 52.8% (103) | -8.4% [-16.9, 0.2] | 0.066 |

CI = confidence interval.

* P < 0.05 (chi-square test of independence: No unused opioids versus any unused opioids).

[a] Percentages calculated excluding missing responses.

[b] Confidence intervals adjusted using Bonferroni correction to control familywise error.

[c] Multiple selections were permitted; percentages may total more than 100%.

less willing to participate (66% vs. 75%, p = 0.026) (Table 2). Among the 28% of respondents who would not participate in the hypothetical program, the most common reasons given were preferring to dispose of the medication on their own (40%), concern about obtaining opioids again in a timely manner (29%), and fear that their pain would return (24%) (Table 3). Compared to respondents with no unused opioids, those with unused opioids at home were more likely to report fear that their pain would return (33% vs. 16%, p = 0.020); other factors were not significantly different between the two groups.

The presence of minors or young adults in the household was not associated with using a secure opioid storage location (Table 4). Compared to respondents who used a locking opioid

**Table 2. Opioid handling information sources and behaviours, overall and by presence of unused opioids at home.**

| Measure | Category | All opioid ever-users, % (count) N = 569 | Unused opioid medication at home | | | P |
|---|---|---|---|---|---|---|
| | | | No unused opioids, % (count) N = 374 | Any unused opioids, % (count) N = 195 | Difference [95% CI] | |
| **Guidance on safe storage of opioids** | | | | | | |
| **Opioid storage information source(s)[c]** | Pharmacist | 35.5% (202) | 37.7% (141) | 31.3% (61) | -6.4% [-14.4, 1.8] | 0.154 |
| | Other healthcare provider | 13.4% (76) | 15.8% (59) | 8.7% (17) | -7.1% [-12.3, -1.4] | 0.026* |
| | Another source | 3.2% (18) | 4.0% (15) | 1.5% (3) | -2.5% [-5.1, 0.6] | 0.178 |
| | I have never received this information | 37.3% (212) | 32.4% (121) | 46.7% (91) | 14.3% [5.8, 22.7] | 0.001* |
| | I don't recall | 15.3% (87) | 15.5% (58) | 14.9% (29) | -0.6% [-6.7, 5.8] | 0.938 |
| **Received storage information from a pharmacist or other healthcare provider** | — | 42.7% (243) | 46.0% (172) | 36.4% (71) | -9.6% [-17.9, -1.1] | 0.035* |
| **Guidance on safe disposal of opioids** | | | | | | |
| **Opioid disposal information source(s)[c]** | Pharmacist | 33.2% (189) | 38.2% (143) | 23.6% (46) | -14.6% [-22.2, -6.7] | <0.001* |
| | Other healthcare provider | 8.4% (48) | 9.6% (36) | 6.2% (12) | -3.5% [-7.8, 1.4] | 0.209 |
| | Another source | 7.6% (43) | 8.6% (32) | 5.6% (11) | -2.9% [-7.1, 1.7] | 0.279 |
| | I have never received this information | 42.9% (244) | 34.8% (130) | 58.5% (114) | 23.7% [15.1, 31.9] | <0.001* |
| | I don't recall | 11.4% (65) | 12.3% (46) | 9.7% (19) | -2.6% [-7.7, 3.0] | 0.441 |
| **Received disposal information from a pharmacist or other healthcare provider** | — | 37.4% (213) | 42.8% (160) | 27.2% (53) | -15.6% [-23.4, -7.4] | <0.001* |
| **Opioid handling behaviors** | | | | | | |
| **Do you have any opioid medication in your household that is no longer being used or is expired?[c]** | Yes, medication prescribed to me | 30.6% (174) | — | 89.2% (174) | — | — |
| | Yes, medication prescribed to someone else | 5.8% (33) | — | 16.9% (33) | — | |
| **How do you store opioid/narcotic medication in your household?[a]** | Locked storage | 9.8% (52) | 13.4% (45) | 3.7% (7) | -9.7% [-14.0, -4.8] | <0.001* |
| | Unlocked storage | 90.2% (476) | 86.6% (292) | 96.3% (184) | 9.7% [4.8, 14.0] | |
| **How do you dispose of your unused opioid medication?[c]** | Flush down the sink or toilet | 7.9% (45) | 8.8% (33) | 6.2% (12) | -2.7% [-7.0, 2.1] | 0.339 |
| | Throw away in garbage | 15.5% (88) | 11.8% (44) | 22.6% (44) | 10.8% [4.2, 17.6] | 0.001* |
| | Mix with undesirable material (e.g. kitty litter, coffee grounds) and throw in garbage | 1.8% (10) | 2.1% (8) | 1.0% (2) | -1.1% [-3.2, 1.4] | 0.533 |
| | Return it to the pharmacy or a community take-back program | 54.1% (308) | 56.7% (212) | 49.2% (96) | -7.5% [-16.0, 1.2] | 0.109 |
| | I would not dispose of it | 13.5% (77) | 9.9% (37) | 20.5% (40) | 10.6% [4.3, 17.1] | <0.001* |
| | Other method | 5.4% (31) | 4.8% (18) | 6.7% (13) | 1.9% [-2.2, 6.3] | 0.465 |
| **Respondent reported a safe disposal method** | — | 55.5% (316) | 58.3% (218) | 50.3% (98) | -8.0% [-16.6, 0.6] | 0.082 |

*(Continued)*

**Table 2.** (Continued)

| Measure | Category | All opioid ever-users, % (count) N = 569 | Unused opioid medication at home | | | P |
| --- | --- | --- | --- | --- | --- | --- |
| | | | No unused opioids, % (count) N = 374 | Any unused opioids, % (count) N = 195 | Difference [95% CI] | |
| **Have you ever shared prescription opioid/ narcotic medication with another person?[b]** | Yes, I have shared medication prescribed to me | 4.6% (26) | 3.2% (12) | 7.2% (14) | 4.0% [-0.9, 9.2] | <0.001* |
| | Yes, I have used medication prescribed to someone else | 2.1% (12) | 1.1% (4) | 4.1% (8) | 3.0% [-0.6, 7.1] | |
| | Yes, I have both shared medication prescribed to me and used medication prescribed to someone else | 1.1% (6) | 0.3% (1) | 2.6% (5) | 2.3% [-0.6, 5.6] | |
| | No | 92.3% (525) | 95.5% (357) | 86.2% (168) | -9.3% [-15.9, -2.9] | |
| **If you had unused opioid/narcotic medication, would you be willing to bring it to your next appointment at the hospital for disposal?[a]** | Yes (versus no) | 71.7% (377) | 75.1% (251) | 65.6% (126) | -9.5% [-17.7, -1.4] | 0.026* |

CI = confidence interval.

* P < 0.05 (chi-square test of independence: No unused opioids versus any unused opioids).

[a] Percentages calculated excluding missing responses.

[b] Confidence intervals adjusted using Bonferroni correction to control familywise error.

[c] Multiple selections were permitted; percentages may total more than 100%.

storage method, respondents who used non-locking storage methods reported similar rates of minors and young adults in the home. Of the 476 respondents who reported storing opioids in an unlocked location, 21% had at least one child (age 0 to 12) and 29% had at least one teenager or young adult (age 13 to 25) in the home.

**Table 3. Reasons for not wanting to participate in an opioid take-back program, overall and by presence of unused opioids at home.**

| Response[a] | All respondents unwilling to participate, % (count) N = 149 | Unused opioid medication at home | | | P |
| --- | --- | --- | --- | --- | --- |
| | | No unused opioids, % (count) N = 83 | Any unused opioids, % (count) N = 66 | Difference [95% CI] | |
| **I am afraid my pain will come back and I'm not willing to be without it** | 23.5% (35) | 15.7% (13) | 33.3% (22) | 17.7% [3.6, 31.1] | 0.020* |
| **If I need it again, I will not be able to get it from a doctor in a timely manner** | 28.9% (43) | 26.5% (22) | 31.8% (21) | 5.3% [-9.3, 19.9] | 0.597 |
| **I do not want to throw it away because I paid for it** | 10.1% (15) | 6.0% (5) | 15.2% (10) | 9.1% [-1.2, 19.4] | 0.118 |
| **My friends or family members may need it** | 3.4% (5) | 3.6% (3) | 3.0% (2) | -0.6% [-6.9, 6.3] | 1.000 |
| **I would prefer to dispose of it myself** | 38.9% (58) | 39.8% (33) | 37.9% (25) | -1.9% [-17.3, 13.8] | 0.948 |
| **Other reason** | 17.4% (26) | 21.7% (18) | 12.1% (8) | -9.6% [-21.1, 2.9] | 0.190 |

CI = confidence interval.

* P < 0.05 (chi-square test of independence: No unused opioids versus any unused opioids).

[a] Multiple selections were permitted; percentages may total more than 100%.

**Table 4. Presence of children, teenagers and/or young adults in the household versus opioid storage method, among current and former opioid users.**

|  | Opioid storage type [a] | | |
| --- | --- | --- | --- |
| Age group | Locked (N = 52) | Not locked (N = 476) | P |
| **Do children, teenagers, or young adults live in your home or visit your home? [b]** | | | |
| No children/teenagers/young adults reported | 57.7% (30) | 56.5% (269) | 0.988 |
| Young children (age 0–6) | 11.5% (6) | 13.0% (62) | 0.932 |
| Older children (age 7–12) | 17.3% (9) | 14.1% (67) | 0.673 |
| Teenagers (age 13–17) | 19.2% (10) | 12.0% (57) | 0.203 |
| Young adults (age 18–25) | 19.2% (10) | 21.8% (104) | 0.796 |
| **Combined age groups:** | | | |
| Derived: Any children (age 0–12) | 23.1% (12) | 20.6% (98) | 0.811 |
| Derived: Any teenagers or young adults (age 13–25) | 32.7% (17) | 28.8% (137) | 0.668 |

[a] 41 respondents who did not report their storage method were excluded.

[b] Multiple selections were permitted; percentages may total more than 100%.

* $P < 0.05$ (chi-square test of independence: Locked storage versus non-locked storage).

## Discussion

This study reveals that many opioid-using orthopaedic patients have unsafe opioid handling practices, and report not receiving instruction on safely managing their prescription opioids from their pharmacist or HCP. In particular, patients with unused opioids in the home reported lower instruction rates, and almost half of this group had minors or young adults in the household. Most orthopaedic patients are willing to participate in a hospital-based opioid disposal program.

We found that orthopaedic patients frequently stored and disposed of opioids incorrectly, consistent with findings from U.S. studies [20]. Only 10% of respondents stored opioids securely, and only 41% of respondents disposed of unused opioids appropriately. A recent systematic review with 810 surgical patients concluded that 77% of patients insecurely stored their opioids and only 9% of patients followed the FDA-recommended disposal methods [21]. Improper storage and retention of unused opioids both increase the risk of opioid misuse, diversion, and accidental poisoning [22,23]. Suboptimal disposal may also contaminate environmental reservoirs [24,25]. Medication take-back programs and coordinated initiatives, such as National Prescription Drug Drop-Off Day in Canada, have been established to promote safe storage and disposal practices and to reduce the number of unused tablets readily available within the community [26]. While promising, these programs tend to have poor uptake and remain in their rudimentary stages of implementation [27,28]. Furthermore, to the best of our knowledge, these programs are seldom promoted in the hospital setting.

While the orthopaedic patients in our study did not explicitly report the reason for their opioid use, it is likely that many of their prescriptions were related to orthopaedic conditions. Short-term opioid use remains essential in the management of postoperative pain or acute orthopaedic injuries [29–34]. Therefore, in order to minimize opioid diversion opportunities, it is imperative that patients receive adequate education regarding safe storage and disposal of opioids. Only a third of patients in our sample recalled receiving instructions on safe opioid handling practices from their pharmacist, and only 10% from other healthcare providers. Other studies in both surgery and emergency departments have found similar low rates of instruction [35–38]. These results suggest a need for both hospital- and community-based

HCPs to address not only prescribing practices [10,11,39], but also patients' opioid handling knowledge (and ultimately, behaviour).

Consistent with other studies, one third of surveyed patients reported having opioids at home that were no longer being used [20,40–42]. We found that these patients were less willing to participate in a hospital-based take-back program compared to those with no unused opioids, citing fear that their pain would return, concerns about untimely access to more opioids, and that they did not want to dispose of medication they paid for. Furthermore, these patients were less likely to report receiving opioid handling education and using safe handling methods, and almost half of the group reported having minors or young adults in the household. This combination of factors may signify a high risk of opioid diversion in this group.

In our study, 8% of current or former opioid users reported sharing their medications with others, and of the 90% who stored their opioids in an unsecured location, 44% reported children, teenagers, or young adults in their household. Previous research indicates that drug misuse is directly linked to the presence of leftover medications in unsecured household cabinets and through sharing unused medications [22,37,43–45]. In fact, 70%-75% of abusers obtained opioids through methods of diversion and only 5% from drug dealers or strangers [46–51]. Consistent with our findings, recent studies have demonstrated that prior instruction in safe opioid handling practices was highly associated with returning medications to a pharmacy, and was the factor most strongly associated with returning medications to a clinician [52–57]. Consequently, it is essential that orthopaedic surgeons, as well as other hospital and community HCPs allocate sufficient time toward educating patients on safe opioid handling practices and the potential danger of opioid diversion. Our subsequent work will aim to determine whether a hospital-based education program is effective for promoting safer opioid handling among patients.

## Limitations

This study had several limitations. Media attention and stigma surrounding opioids may have contributed to a social desirability bias, leading participants to underreport opioid use, retention of unused opioids, and unsafe handling practices. An anonymous survey and survey return mechanism were used to minimize this effect. Recall bias may also lead participants to misreport prior education on opioid handling. Second, the voluntary nature of the questionnaire creates a risk of selection bias. The representativeness of the sample cannot be confirmed, as the number of patients who declined to participate was not tracked. Although the sample's age and gender distribution is similar to that of the general orthopaedic population [58], the study was conducted in a single large, urban, tertiary care centre which may limit the generalizability of the findings. Lastly, the study data was collected at a single time point; patients' opioid handling practices may have changed with increased awareness of the opioid epidemic.

## Conclusion

Patients attending an ambulatory Orthopedic clinic who have been prescribed opioids commonly report possession of unused opioids and unsafe opioid handling practices. As well, relatively few patients prescribed opioids report receiving education on safe opioid handling practices from their pharmacist, and even fewer from other healthcare providers. Our findings reveal a significant opportunity for hospital-driven opioid stewardship. While overprescribing of opioids remains a critical issue, our study suggests a concurrent need to address patient knowledge and behavior regarding opioid handling in order to reduce the risk of opioid diversion in the households of opioid-using patients.

## Supporting information

**S1 Table. Comparative analysis of opioid-using respondents by receipt of opioid storage and disposal instruction.**
(PDF)

**S1 Appendix. Public safety survey.**
(PDF)

## Acknowledgments

The University Health Network Division of Orthopaedic Surgery comprises the following members: J. Chahal, J. R. Davey, R. Gandhi, J. Lau, T. Leroux, S. Lewis, N. Mahomed, K. W. Marshall, D. Ogilvie-Harris, Y. R. Rampersaud (corresponding author), K. Syed, C. Veillette, and M. Zywiel.

## Author Contributions

**Conceptualization:** Kala Sundararajan, Mayilee Canizares, J. Denise Power, Anthony V. Perruccio, Angela Sarro, Luis Montoya, Y. Raja Rampersaud.

**Data curation:** Kala Sundararajan.

**Formal analysis:** Kala Sundararajan.

**Funding acquisition:** Y. Raja Rampersaud.

**Investigation:** Kala Sundararajan, Angela Sarro, Luis Montoya.

**Methodology:** Kala Sundararajan, Mayilee Canizares, J. Denise Power, Anthony V. Perruccio, Angela Sarro, Y. Raja Rampersaud.

**Project administration:** Kala Sundararajan.

**Resources:** Y. Raja Rampersaud.

**Supervision:** Y. Raja Rampersaud.

**Writing – original draft:** Kala Sundararajan, Prabjit Ajrawat, Y. Raja Rampersaud.

**Writing – review & editing:** Kala Sundararajan, Prabjit Ajrawat, Mayilee Canizares, J. Denise Power, Anthony V. Perruccio, Angela Sarro, Luis Montoya, Y. Raja Rampersaud.

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
