## [Decision Letter · Decision Letter 0]

21 Jun 2021

PONE-D-21-17272

The potential for diversion of prescribed opioids among orthopaedic patients: results of an anonymous patient survey

PLOS ONE

Dear Dr. Rampersaud,

Thank you for submitting your manuscript to PLOS ONE. After careful consideration, we feel that it has merit but does not fully meet PLOS ONE’s publication criteria as it currently stands. Therefore, we invite you to submit a revised version of the manuscript that addresses the points raised during the review process.

We look forward to receiving your revised manuscript.

Kind regards,

Vijayaprakash Suppiah, PhD

Academic Editor

PLOS ONE

Journal Requirements:

1. Please ensure that your manuscript meets PLOS ONE's style requirements, including those for file naming. The PLOS ONE style templates can be found athttps://journals.plos.org/plosone/s/file?id=wjVg/PLOSOne_formatting_sample_main_body.pdf and https://journals.plos.org/plosone/s/file?id=ba62/PLOSOne_formatting_sample_title_authors_affiliations.pdf

'I have read the journal's policy and the authors of this manuscript have the following

competing interests: Y. Raja Rampersaud has received royalties from Medtronic and

holds investments in Arthur Health Corporation.

All other co-authors have no competing interests to report.'

Additional Editor Comments (if provided):

Reviewers' comments:

Reviewer's Responses to Questions

**Comments to the Author**

1. Is the manuscript technically sound, and do the data support the conclusions?

Reviewer #1: Yes

Reviewer #2: Yes

2. Has the statistical analysis been performed appropriately and rigorously? 

Reviewer #1: I Don't Know

Reviewer #2: Yes

3. Have the authors made all data underlying the findings in their manuscript fully available?

Reviewer #1: No

Reviewer #2: Yes

4. Is the manuscript presented in an intelligible fashion and written in standard English?

Reviewer #1: Yes

Reviewer #2: Yes

5. Review Comments to the Author

Reviewer #1: Dear authors,

I think this analysis in your article is an important finding, considering the opioid epidemic in Canada and the numbers you raised for hospitalization.

The abstract clearly summarizes the study and is adequate.

The introduction is adequate.

For the method part, you write that existing studies defined safe storage location as protected by a lock. So why did you consider both latch and lock as safe for your study? A Latch for me is not really safe, as beginning with small children or anyone else getting to that room will be able to open it. Does your data change when you only consider a lock as a safe storage location?

What would be interesting is establishing a education program e.g. in your clinic and to the measurement again, to see if your hypothesis that education in safe storage and so on has any effect on return rates or safe storage. In the same way it might be interesting to investigate, as you say, if patient handling with opioids has changed by the awareness of the opiod epidemic und media attention.

You state that patients with unused opioids at home have significantly less likely received storage information or disposal instructions. The questions that raises to me is on the other hand is if patients who have received storage information and disposal instructions are more likely to use a safe storage location, have higher rates in returning opioids, have less unused opioids at home and so on. This is the questions to answer if one wants to see if education on opiod use has any effect. This is somehow the conclusion of your study, that more education is needed, but is there any proof that education changes the rates of safe opiod handling? Can you answer that out of your data?

Reviewer #2: - Abstract/ Results: consider adding a denomination, e.g. '(past) prescription opioid use' to better describe the study population in summary.

- Table 2 is an extensive table, leading to reduced legibility. Please provide some separation between parts of the table, either within this table, or by creating multiple tables.

- Discussion, second paragraph: limitations were described in profound detail. However, it could be of interest to add some background information on chronic opioid use in relation to the study population. E.g.: J Pain. 2017 November ; 18(11): 1374–1383

6. PLOS authors have the option to publish the peer review history of their article (what does this mean?). If published, this will include your full peer review and any attached files.

Reviewer #1: No

Reviewer #2: **Yes: **FGAM van Haren

---

## [Author Response · Author response to Decision Letter 0]

5 Aug 2021

Please see attached Response to Reviewers document.

---

## [Decision Letter · Decision Letter 1]

16 Aug 2021

The potential for diversion of prescribed opioids among orthopaedic patients: results of an anonymous patient survey

PONE-D-21-17272R1

Dear Dr. Rampersaud,

We’re pleased to inform you that your manuscript has been judged scientifically suitable for publication and will be formally accepted for publication once it meets all outstanding technical requirements.

Kind regards,

Kingston Rajiah

Academic Editor

PLOS ONE

Reviewers' comments:

Reviewer's Responses to Questions

**Comments to the Author**

1. If the authors have adequately addressed your comments raised in a previous round of review and you feel that this manuscript is now acceptable for publication, you may indicate that here to bypass the “Comments to the Author” section, enter your conflict of interest statement in the “Confidential to Editor” section, and submit your "Accept" recommendation.

Reviewer #1: All comments have been addressed

Reviewer #2: All comments have been addressed

2. Is the manuscript technically sound, and do the data support the conclusions?

Reviewer #1: Yes

Reviewer #2: Yes

3. Has the statistical analysis been performed appropriately and rigorously? 

Reviewer #1: Yes

Reviewer #2: Yes

4. Have the authors made all data underlying the findings in their manuscript fully available?

Reviewer #1: Yes

Reviewer #2: Yes

5. Is the manuscript presented in an intelligible fashion and written in standard English?

Reviewer #1: Yes

Reviewer #2: Yes

6. Review Comments to the Author

Reviewer #1: All points that have been raised after the first submission have been answered and corrected in the new submission, good work.

Reviewer #2: Already of good quality, comment have been adequately addressed. Already of good quality, comment have been adequately addressed.

7. PLOS authors have the option to publish the peer review history of their article (what does this mean?). If published, this will include your full peer review and any attached files.

Reviewer #1: No

Reviewer #2: No

---

## [Editor Report · Acceptance letter]

18 Aug 2021

PONE-D-21-17272R1 

The potential for diversion of prescribed opioids among orthopaedic patients: results of an anonymous patient survey 

Dear Dr. Rampersaud:

I'm pleased to inform you that your manuscript has been deemed suitable for publication in PLOS ONE. Congratulations! Your manuscript is now with our production department. 

Kind regards, 

on behalf of

Dr. Kingston Rajiah 

Academic Editor

PLOS ONE